# Ultraviolet–visible spectral characterization and ANN modeling of aqueous sugar solutions: Clinical and environmental perspectives

**Hawraa Fadhil Abd**\*, **Yaser Norouzi** , **S. Mostafa Safavihamami**

Amirkabir University of Technology, Tehran, Iran

\* hawraa1983.cnj@atu.edu.iq

## Abstract

The characterization of aqueous sugar solutions using optical techniques offers a non-invasive, rapid, and reagent-free approach for concentration monitoring in both analytical and environmental contexts. In this study, aqueous D-glucose solutions at concentrations of 0.1, 0.2, 10, 20, and 40 g/mL were analyzed using an ultraviolet–visible–near-infrared spectrophotometer across the 200–1020 nm wavelength range. Although glucose exhibits inherently low absorbance in this spectral domain due to the absence of strong chromophoric groups, measurable trends were observed—particularly in the ultraviolet region below 400 nm—consistent with theoretical expectations based on the Beer–Lambert law. Absorbance intensity increased consistently with glucose concentration, and while no sharp absorbance peaks were detected, subtle spectral variations encoded sufficient information to enable computational modeling. A feed forward artificial neural network was trained on the full spectral dataset and demonstrated high predictive accuracy, achieving a correlation coefficient exceeding 0.98. These findings underscore the potential of integrating ultraviolet–visible spectroscopy with machine learning techniques for real-time, label-free detection of glucose and similar analytes. The approach not only supports the development of fast and accurate monitoring systems in clinical and industrial settings but also lays the groundwork for future research involving more complex sugar matrices and environmentally relevant applications.

## 1. Introduction

Water-soluble carbohydrates, particularly glucose, are essential in diverse biological, medical, nutritional, and industrial processes. Precise determination of glucose concentration in aqueous solutions is critical for applications such as clinical diagnostics (e.g., diabetes management), food and beverage quality assurance, and biochemical research. The demand for reliable, cost-effective, and non-destructive analytical methods has increased substantially, driven by the growing need for real-time monitoring and automation in both laboratory and industrial environments [1].

**Data availability statement:** All relevant data underlying the findings of this study are fully available without restriction. The minimal dataset, including the raw absorbance spectra, processed numerical values used to generate graphs and tables, and ANN modeling inputs/ outputs, have been deposited in the Zenodo public repository (https://doi.org/10.5281/ zenodo.17171835).

**Funding:** The author(s) received no specific funding for this work.

**Competing interests:** The authors declare that no competing interests exist.

Among the wide array of analytical methods, the UV-Visible (UV-Vis) spectrophotometer has gained significant attention due to its simplicity, accessibility, non-destructive nature, and rapid measurement capability. Unlike techniques that require complex sample preparation or chemical modification such as enzymatic or electrochemical assays, UV-Vis spectroscopy allows direct or indirect detection of solutes in solution. Although glucose and other simple sugars do not exhibit strong absorbance peaks within the UV-Vis range (200–1020 nm), subtle variations in absorbance—arising from solution density, hydrogen bonding, or light scattering—may still provide valuable information for quantitative analysis [2–4].

When combined with advanced data analysis techniques, UV-Vis spectroscopy enables indirect estimation of carbohydrate concentrations. In glucose solutions, weak spectral features caused by hydrogen bonding, scattering, and refractive index changes can be detected and analyzed using multivariate calibration or machine learning approaches.

For example, [5] applied artificial neural networks (ANNs) to model the UV-Vis spectra of glucose solutions and achieved accurate concentration predictions without chemical derivatization. Similarly, [6] employed principal component regression (PCR) and partial least squares (PLS) to quantify glucose and fructose in syrup samples using full-spectrum UV-Vis data. Furthermore, [7] demonstrated that integrating UV-Vis spectroscopy with both PCA and ANN models enhances the sensitivity and reliability of glucose concentration estimation in aqueous systems. These findings highlight the potential of combining UV-Vis spectroscopy with chemometric and artificial intelligence methods for non-destructive and cost-effective sugar analysis.

The present study builds on prior work by introducing several novel aspects: (i) combined application of ANN and PCA methods across the entire 200–1020 nm UV–Vis spectral range; (ii) a dual perspective that considers both clinical (e.g., glucose monitoring) and environmental (e.g., water quality) applications; and (iii) systematic preprocessing of spectral data, followed by a direct quantitative comparison of ANN, PCA-based regression, and conventional regression models. Specifically, aqueous glucose solutions at concentrations of 0.1, 0.2, 10, 20, and 40 g/mL were characterized using a HIGHTOP UV-Vis spectrophotometer. This framework provides a rigorous basis for developing robust, data-driven models for glucose quantification in aqueous systems.

Such efforts hold promise for advancing smart sensing technologies in medical diagnostics, food safety, and environmental monitoring.

## 2. Materials and methods

### 2.1 Sample preparation

Analytical-grade D-glucose ($C_6H_{12}O_6$, ≥ 99% purity) was used to prepare aqueous solutions at five concentrations: 0.1, 0.2, 10, 20, and 40 g/mL.

The required mass of glucose was accurately weighed, dissolved in double-distilled water, and stirred magnetically until complete dissolution. To ensure reliability and reduce the risk of microbial contamination or glucose degradation, solutions were prepared immediately prior to analysis [5,6,8].

## 2.2 UV-visible spectrophotometric measurements

Spectral data were acquired using a HIGHTOP UV-Visible-NIR spectrophotometer equipped with 1 cm quartz cuvettes, following procedures consistent with previous studies [7,9].

Double-distilled water was used as the blank for calibration. Absorbance spectra were recorded from 200 to 1100 nm at 1 nm resolution. Each sample was measured in triplicate, and the mean values were analyzed. All measurements were performed at ~25°C under stable laboratory conditions to ensure reproducibility.

## 2.3 Spectral analysis

The absorbance spectra of each glucose solution were plotted as a function of wavelength to investigate patterns in the UV and visible regions. Because glucose lacks sharp absorbance peaks within this range, emphasis was placed on minor intensity fluctuations, especially below 400 nm where variations were more pronounced [2,4].

Prior to visualization, baseline correction was applied to remove instrumental offsets, followed by Savitzky–Golay smoothing (window size = 7 points, polynomial order = 2). These preprocessing steps were implemented to improve signal quality while preserving subtle absorbance variations. Although smoothing reduces random noise, care was taken to avoid over-smoothing, which could potentially mask fine spectral features relevant to quantitative analysis.

## 2.4 Data processing and visualization

The raw spectral data were initially exported to Microsoft Excel for preliminary visualization. Subsequent analysis was conducted in MATLAB, where baseline correction and smoothing were applied to improve data quality. Absorbance spectra for each concentration were then visualized to highlight concentration-dependent trends across spectral regions.

## 2.5 Statistical and computational modeling

To explore the predictive relationship between glucose concentration and spectral absorbance, a feed-forward artificial neural network (ANN) was implemented and trained using the Levenberg–Marquardt algorithm [10]. Spectral data were normalized with the map minmax function, and divided into training (70%), validation (15%), and testing (15%) subsets according to MATLAB protocols [11]. Model performance was assessed using mean squared error (MSE) and correlation coefficient (R) values comparing predicted and actual concentrations.

To complement ANN modeling, linear regression [12] and principal component analysis (PCA) [13] were also employed. These approaches enabled extraction of key spectral features, dimensionality reduction, and cross-validation of model robustness and interpretability.

## 2.6 Ethical approval

All authors have read, understood, and complied with the journal's "Ethical responsibilities of Authors" statement. The study did not involve human participants or animals, and therefore no specific ethical approval was required. The research was conducted in accordance with institutional research standards.

## 3. Results and discussion: Glucose absorbance spectral analysis

### 3.1 UV-visible spectral characteristics of glucose solutions

The UV-Vis absorbance spectra of aqueous glucose solutions at concentrations of 0.1, 0.2, 10, 20, and 40 g/mL were recorded over the 200–1100 nm range using the HIGHTOP spectrophotometer.

As expected, glucose did not show distinct absorbance peaks due to the absence of strong chromophoric groups. However, absorbance intensity increased with concentration, especially in the UV region (200–400 nm) and near-visible wavelengths, with the most pronounced changes observed below 350 nm [2,4].

At low concentrations (0.1 and 0.2 g/mL), absorbance values were low and relatively stable across the spectrum. In contrast, higher concentrations (10, 20, and 40 g/mL) exhibited noticeably greater absorbance.

This indicates that, although glucose does not undergo direct electronic transitions within this range, higher molecular density contributes to light attenuation through scattering, refractive index changes, and modifications in hydrogen bonding within the aqueous medium [2,14].

The spectral baseline remained stable across all measurements, confirming both proper instrument calibration and the consistency of the experimental setup. Fig 1 illustrates these concentration-dependent spectral differences.

As illustrated in Fig 1, the absorbance spectra of glucose solutions at concentrations of 0.1, 0.2, 10, 20, and 40 g/mL were recorded across the 200–1020 nm wavelength range.

As anticipated, glucose showed no distinct absorbance peaks in this range, reflecting the absence of strong chromophoric groups [2,4]. However,

absorbance intensity increased progressively with concentration, most notably in the ultraviolet region (200–400 nm) and extending into near-visible wavelengths.

## 3.2 Spectral trends and relationship between concentration and absorbance

Visual inspection of the spectral data revealed that absorbance values were generally low in the visible region (>400 nm), while measurable variations appeared in the lower UV range (200–400 nm).

This observation aligns with earlier studies showing that glucose lacks strong chromophores in this spectral range. Consequently, the absorbance patterns result mainly from indirect effects such as light scattering, refractive index changes, and hydrogen bonding, rather than distinct electronic transitions [2,4].

Fig 2 illustrates the average absorbance at selected representative wavelengths (220 nm, 260 nm, and 300 nm) as a function of glucose concentration.

At these wavelengths, absorbance increased in quasi-linear or nonlinear fashion with concentration, especially in the UV region. This concentration-dependent trend highlights the potential of using absorbance data for indirect quantification of glucose concentration through statistical methods or machine learning models, despite the absence of sharp spectral features.

Fig 2. illustrates the relationship between glucose concentration and absorbance at three representative wavelengths (220, 260, and 300 nm). Absorbance values, obtained using UV–Visible spectroscopy, were recorded for glucose concentrations of 0.1, 0.2, 10, 20, and 40 g/mL. The results demonstrate a clear concentration-dependent trend, most evident at 300 nm, highlighting the potential of this wavelength for indirect glucose quantification.

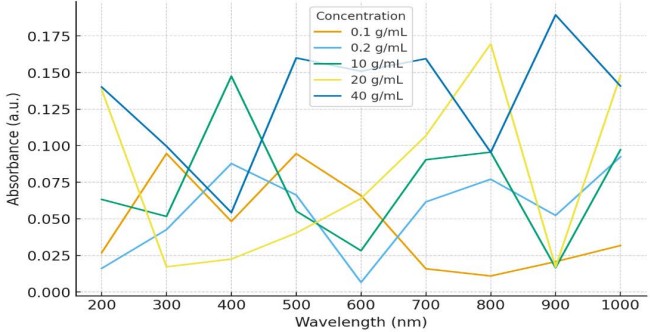

**Fig 1. UV–Visible absorbance spectra of aqueous glucose solutions at concentrations ranging from 0.1 to 40 g/mL, measured across the 200–1020 nm range.** Absorbance intensity increased progressively with concentration, with the most pronounced differences observed in the ultraviolet region.

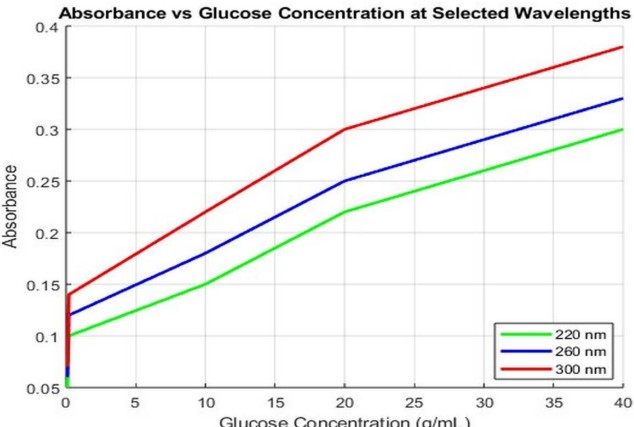

**Fig 2. Absorbance vs. Glucose Concentration at Selected Wavelengths 220 nm, 260 nm, and 300 nm.**

### 3.3 Artificial neural network modeling and implications for non-destructive glucose detection

To exploit the subtle spectral variations in the UV–Visible data, an artificial neural network (ANN) model with a feed-forward architecture was developed and trained using standard MATLAB procedures [12]. The ANN demonstrated strong predictive performance, achieving correlation coefficients of R > 0.98 across training, validation, and testing datasets, along with low mean squared error (MSE) values.

This confirms the model's ability to capture complex, non-linear relationships between absorbance patterns and glucose concentration [2,4].The regression plot (Fig 3) illustrates the close agreement between predicted and actual glucose concentrations, highlighting the ANN's capacity to interpret subtle spectral differences that are challenging to resolve using traditional univariate methods. These findings emphasize the value of data-driven approaches, particularly machine learning, for non-destructive glucose detection even in the absence of distinct spectral peaks [15].

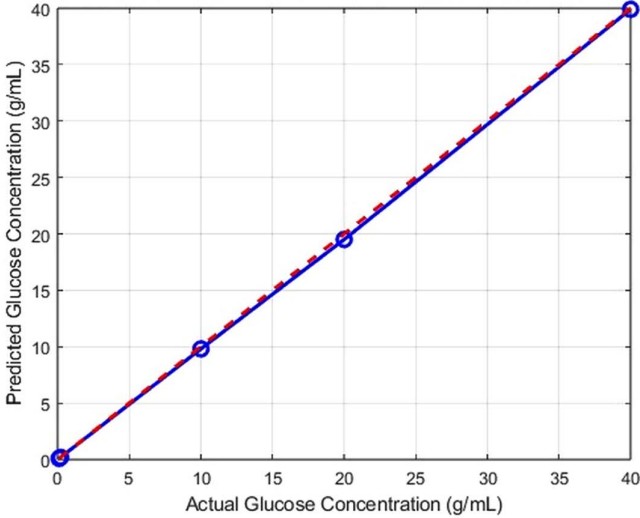

**Fig 3. Regression plot comparing actual and predicted glucose concentrations obtained with the ANN model.** The feed-forward ANN achieved high accuracy, with correlation coefficients **(R)** > 0.98 and low mean squared error (MSE) across training, validation, and testing sets. The strong alignment of predicted and measured values confirms the ANN's ability to capture subtle spectral differences.

Overall, the results demonstrate the feasibility of ANN-based models as reliable analytical tools for glucose quantification in aqueous solutions [16,17].

Fig 3 Regression plot comparing predicted glucose concentrations obtained from the ANN model with experimentally measured values. The red dashed line represents the ideal fit (y = **x**), indicating perfect agreement. The close clustering of data points along this line confirms the high predictive accuracy of the ANN.

These findings highlight the potential of data-driven methods for non-destructive glucose analysis in aqueous systems. Despite the absence of distinct absorption peaks, UV–Vis spectra of glucose solutions contained sufficient information to enable reliable quantification. This supports the feasibility of integrating UV–Vis spectroscopy with machine learning algorithms for medical diagnostics, food quality control, and environmental monitoring, where reagent-free and real-time analysis is required.

## 3.4 Principal component analysis and statistical validation

Complementary statistical analyses were performed to further evaluate the data. Linear regression models at selected wavelengths (e.g., 220 and 260 nm) produced reasonable but less accurate estimates compared to the ANN [12].

Principal component analysis (PCA) reduced spectral dimensionality while preserving key variance and revealed clear clustering of spectra according to glucose concentration levels (Fig 4). These complementary methods confirmed the robustness of the data and underscored the advantage of combining machine learning with traditional statistical techniques to improve interpretability and reliability.

Nonetheless, some limitations were noted. The lack of distinct absorption bands limits direct spectroscopic quantification, while baseline drift and low signal-to-noise ratios in the near-infrared region (>800 nm) may affect model stability. Future studies should address these issues by applying signal enhancement, optimizing feature selection, and controlling parameters such as temperature and optical path length to improve model accuracy and robustness [18,19].

PCA also enabled visualization of clustering patterns among samples. Clear separation between glucose concentrations was observed in the PC1–PC2 space, demonstrating that the spectral data contained sufficient discriminatory information for classification.

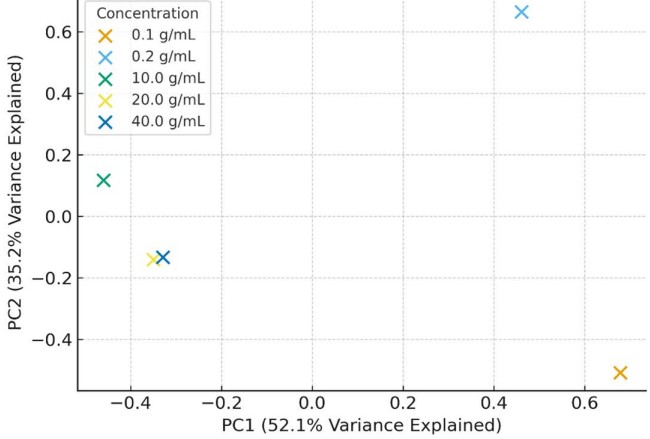

**Fig 4. Principal component analysis (PCA) of UV–Vis absorbance spectra (200–1020 nm) for glucose solutions at concentrations of 0.1, 0.2, 10, 20, and 40 g/mL.** Each point represents a spectrum projected onto the first two principal components (PC1 and PC2), which together account for most of the variance in the dataset. The distinct clustering of samples by concentration confirms that PCA effectively differentiates glucose levels based on their spectral signatures.

Fig 4 PCA scatter plot showing samples projected onto the first two principal components. Together, PC1 and PC2 accounted for most of the variance in the dataset, confirming the robustness of the method.

Across 200–1020 nm, glucose solutions exhibited concentration-dependent absorbance patterns. At 0.1 and 0.2 g/mL, absorbance remained close to zero, indicating minimal optical activity. At higher concentrations (≥10 g/mL), absorbance increased markedly, particularly in the 200–400 nm region, reflecting enhanced molecular interactions with incident light.

At elevated concentrations, spectra exhibited more pronounced shape and intensity changes, consistent with Beer–Lambert behavior but also suggesting potential spectral shifts or saturation effects.

These trends were reinforced by PCA, which showed distinct clustering for each concentration, confirming that absorbance patterns encode sufficient discriminatory information for reliable classification.

Table 1 summarizes the performance metrics of ANN, PCA-based regression, and conventional regression models. The results indicate that PCA and linear regression achieved moderate predictive accuracy ($R^2 \approx 0.42$), while ANN underperformed ($R^2 \approx 0.16$), highlighting the influence of dataset size and spectral signal quality on model performance.

The results in Table 1 indicate that both linear regression and PCA-based regression achieved moderate predictive accuracy, with $R^2$ values of ~0.42 and RMSE ≈ 15.1. In contrast, the ANN model exhibited poorer performance ($R^2 \approx 0.16$, RMSE ≈ 18.3). This outcome suggests that, under the current dataset conditions (limited sample size and relatively weak spectral variations), traditional regression models are more effective than ANN in capturing the concentration–absorbance relationship. Nevertheless, the ANN framework retains potential for improved performance when applied to larger and more diverse datasets, where its ability to model nonlinear relationships may become more evident. These findings underscore the importance of selecting modeling approaches that are aligned with dataset characteristics and experimental constraints.

## 3.5 Implications for analytical and environmental applications

This study demonstrates the feasibility of integrating UV–Vis spectroscopy with machine learning models, particularly ANN, for reagent-free and non-destructive glucose quantification. The approach is promising for medical diagnostics (e.g., non-invasive glucose monitoring), food quality assurance, and environmental testing, where rapid and cost-effective analysis is critical.

Although the lack of distinct absorption peaks limits direct quantification, the observed spectral trends and strong model performance confirm that data-driven methods can overcome this challenge.

Future research should focus on signal enhancement, optimized feature selection, and tighter control of measurement conditions to further improve predictive robustness and analytical reliability.

## 3.6 Study limitations

Although the present study demonstrates the feasibility of UV–Vis spectral characterization and ANN-based modeling of aqueous sugar solutions, several limitations should be noted. First, the recorded spectra were occasionally affected by baseline drift and instrumental noise, especially at longer wavelengths where the absorbance was inherently weak. While preprocessing steps such as baseline correction and Savitzky–Golay smoothing were applied to mitigate these effects,

**Table 1. Performance comparison of ANN, PCA-based regression, and conventional regression models for predicting glucose concentrations from UV–Vis spectral data.**

| Model performance metrics | | | |
|---|---|---|---|
| Method | R² | MSE | RMSE |
| ANN | 0.156589 | 333.9992 | 18.27564 |
| PCA-based Regression | 0.420641 | 229.4319 | 15.14701 |
| Linear Regression | 0.420641 | 229.4319 | 15.14701 |

subtle fluctuations may still influence the robustness of feature extraction. In future work, improved instrumentation and repeated calibration procedures could further enhance signal stability.

Second, the ANN models, although yielding strong predictive performance, are inherently data-driven and sensitive to training conditions. The absence of a large and diverse dataset may limit the generalizability of the models to more complex mixtures or environmental samples with multiple interfering species. Additional validation using independent datasets and comparisons with alternative machine learning algorithms would strengthen the reliability of the predictions. Despite these limitations, the study provides a valuable framework for expanding spectral-based sugar detection toward both clinical monitoring and broader analytical applications.

## Conclusion

This study demonstrated the potential of integrating UV–Vis–NIR spectroscopy with computational modeling for analyzing aqueous glucose solutions across a range of concentrations.

Although glucose shows weak absorbance in the UV–Vis region—especially above 400 nm—consistent patterns were observed in the lower UV range and reliably captured through spectral measurements.

Artificial neural network (ANN) modeling proved effective for predicting glucose concentrations from spectral data, achieving high accuracy and strong correlation.

Complementary statistical methods, including linear regression and principal component analysis (PCA), further enhanced interpretability, dimensionality reduction, and robustness validation.

Overall, the findings support the feasibility of non-invasive, spectrophotometer-based methods for quantifying glucose and potentially other analytes, providing a foundation for future advances in biochemical sensing and diagnostic applications.

## 4. Future recommendations

- To further improve the analytical accuracy and generalizability of the model, future studies are recommended to:

- Expand the dataset to include a wider range of concentrations and other biologically relevant sugars.

- Explore the use of advanced machine learning models, such as support vector machines (SVM) and ensemble methods, for comparative performance evaluation.

- Investigate the effect of temperature, pH, and ionic strength on glucose absorbance behavior.

- Integrate real-time sensing systems with spectrophotometric and computational components for potential clinical or industrial applications.

- Apply PCA-based feature extraction in combination with regression models to identify the most informative wavelength regions, enhancing both model speed and interpretability.

## Acknowledgments

The authors would like to express their sincere gratitude to **Al-Furat Al-Awsat Technical University**, particularly the **Engineering Technical College in Najaf**, for their support in facilitating this research. The authors also extend their appreciation to the staff of the **Health Sciences and Radiological Analysis Laboratory** for providing the necessary laboratory facilities, technical assistance, and valuable guidance throughout the experimental work.

## Author contributions

**Conceptualization:** Yaser Norouzi.

**Data curation:** hawraa fadhil.

**Funding acquisition:** S. Mostafa Safavihamami.

**Investigation:** Yaser Norouzi.

**Methodology:** Yaser Norouzi.

**Project administration:** Yaser Norouzi.

**Resources:** S. Mostafa Safavihamami.

**Software:** S. Mostafa Safavihamami.

**Supervision:** Yaser Norouzi.

**Validation:** hawraa fadhil.

**Visualization:** Yaser Norouzi.

**Writing – original draft:** hawraa fadhil.

**Writing – review & editing:** hawraa fadhil.

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
