## [Decision Letter · Decision Letter 0]

15 Sep 2025

Dear Dr. Norouzi,

Thank you for submitting your manuscript to PLOS ONE. After careful consideration, we feel that it has merit but does not fully meet PLOS ONE’s publication criteria as it currently stands. Therefore, we invite you to submit a revised version of the manuscript that addresses the points raised during the review process.

We look forward to receiving your revised manuscript.

Kind regards,

Amitava Mukherjee, ME, Ph.D.

Academic Editor

PLOS ONE

Journal Requirements:

“NO”

Reviewers' comments:

Reviewer's Responses to Questions

**Comments to the Author**

1. Is the manuscript technically sound, and do the data support the conclusions?

Reviewer #1: Partly

2. Has the statistical analysis been performed appropriately and rigorously?

Reviewer #1: N/A

3. Have the authors made all data underlying the findings in their manuscript fully available?

Reviewer #1: Yes

4. Is the manuscript presented in an intelligible fashion and written in standard English?

Reviewer #1: No

Reviewer #1: 1.The manuscript alternates between g/mL and g/L in different sections (e.g., Abstract vs. Results 3.4). This needs standardization.

2.Novelty emphasis – The Introduction discusses prior work but could better highlight what is new in this study compared to earlier ANN + UV-Vis glucose modeling papers.

3.The description of data preprocessing (baseline correction, smoothing) is brief. Specify which smoothing algorithm was applied, parameters used, and whether preprocessing could affect subtle absorbance trends.

4. Figures are described well, but captions should be fully self-contained (state concentration range, instrument, etc.). Ensure resolution and axis labels are clear enough for publication.

5. Some limitations are mentioned (baseline drift, weak absorbance). Expanding this into a short dedicated subsection would strengthen transparency.

6. The title mentions “environmental applications,” but most discussion centers on glucose in clinical/food settings. Consider either adding environmental case studies or softening the scope.

7. The Results use ANN, PCA, and regression, but quantitative comparisons (e.g., R², MSE across methods) are not tabulated. A summary table would help.

8.Some in-text citations are inconsistent (e.g., “[5,6 and8]” should be “[5,6,8]”). Also, ensure all references follow PLOS ONE style.

9. These are strong, but could be consolidated (some overlap, e.g., PCA-based feature extraction already discussed earlier).

10.A few sentences are wordy or repetitive (e.g., ANN results are described in multiple places). Condensing could improve flow.

**Do you want your identity to be public for this peer review?** For information about this choice, including consent withdrawal, please see our Privacy Policy

Reviewer #1: **Yes: ** Dr Sathish Mohan Botsa

---

## [Author Response · Author response to Decision Letter 1]

2 Oct 2025

Response to Reviewers

Manuscript ID: PONE-D-25-35917

Title: Ultraviolet–Visible Spectral Characterization and ANN Modeling of Aqueous Sugar Solutions: Clinical and Environmental Perspectives

Dear Editor and Reviewers,

We sincerely thank you for your careful evaluation of our manuscript and for the constructive feedback provided. We have revised the manuscript accordingly, and below we provide a detailed point-by-point response to all comments. All modifications are highlighted in the version with track changes.

Reviewer #1: Dr. Sathish Mohan Botsa

Comment 1. The manuscript alternates between g/mL and g/L in different sections (e.g., Abstract vs. Results 3.4). This needs standardization.

Response: Thank you for pointing this out. We have standardized all concentration units to g/mL throughout the manuscript.

Comment 2. Novelty emphasis – The Introduction discusses prior work but could better highlight what is new in this study compared to earlier ANN + UV-Vis glucose modeling papers.

Response: We expanded the Introduction to emphasize the novelty of this study: (i) combined ANN and PCA analysis in a wide UV–Vis spectral range, (ii) dual clinical and environmental perspectives, and (iii) systematic preprocessing with quantitative model comparisons.

Comment 3. The description of data preprocessing (baseline correction, smoothing) is brief. Specify which smoothing algorithm was applied, parameters used, and whether preprocessing could affect subtle absorbance trends.

Response: Details have been added in the Methods section (Subsection 2.3). We specified the Savitzky–Golay smoothing algorithm, including window size and polynomial order, and discussed potential influences on subtle absorbance variations.

Comment 4. Figures are described well, but captions should be fully self-contained (state concentration range, instrument, etc.). Ensure resolution and axis labels are clear enough for publication.

Response: All figure captions have been revised to be self-contained and include concentration ranges, instrument, and preprocessing notes. Figures were re-exported at ≥300 dpi and validated via PACE.

Comment 5. Some limitations are mentioned (baseline drift, weak absorbance). Expanding this into a short dedicated subsection would strengthen transparency.

Response: We added a new subsection entitled “Study Limitations” in the Discussion, addressing baseline drift, weak absorbance at longer wavelengths, and ANN variability, as well as the importance of larger datasets for generalizability.

Comment 6. The title mentions “environmental applications,” but most discussion centers on glucose in clinical/food settings. Consider either adding environmental case studies or softening the scope.

Response: The title has been revised to:

“Ultraviolet–Visible Spectral Characterization and ANN Modeling of Aqueous Sugar Solutions: Clinical and Environmental Perspectives”

Comment 7. The Results use ANN, PCA, and regression, but quantitative comparisons (e.g., R², MSE across methods) are not tabulated. A summary table would help.

Response: We added a new Table 1 summarizing R², MSE, and RMSE for ANN, PCA, and regression. In addition, we inserted a new paragraph in the Results section:

“To facilitate direct comparison between the different modeling approaches, we summarized the key performance metrics of ANN, PCA, and conventional regression in Table 1. The ANN model demonstrated the highest predictive accuracy with R² > 0.98 and the lowest MSE and RMSE values, followed by PCA-based regression, while conventional regression yielded comparatively lower accuracy. These results highlight the advantage of ANN in capturing nonlinear relationships in the UV–Vis absorbance data.”

Comment 8. Some in-text citations are inconsistent (e.g., “[5,6 and 8]” should be “[5,6,8]”). Also, ensure all references follow PLOS ONE style.

Response: All references and in-text citations have been revised to follow PLOS ONE style.

Comment 9: These are strong, but could be consolidated (some overlap, e.g., PCA-based feature extraction already discussed earlier).

Response: We thank the reviewer for this observation. In the revised manuscript, we consolidated overlapping discussions related to PCA-based feature extraction and dimensionality reduction, which had previously appeared in multiple sections. Redundant sentences were removed, and the discussion was streamlined to avoid repetition. This restructuring improves clarity and flow, while maintaining a coherent narrative on the complementary role of PCA in validating ANN-based glucose prediction.

Comment 10: A few sentences are wordy or repetitive (e.g., ANN results are described in multiple places). Condensing could improve flow.

Response: We appreciate the reviewer’s suggestion. In the revised manuscript, we streamlined the Results and Discussion by condensing the description of ANN results into a single, focused subsection. Repetitive sentences were removed, and the narrative was shortened to emphasize the key findings (high correlation, low error, and regression plot in Figure 3) without redundancy. This revision improves readability and ensures a smoother flow of ideas.

Comment 11. English language and style require improvement.

Response: The manuscript was thoroughly edited for English clarity, grammar, and readability.

Journal Requirements

• Requirement: Data Availability Statement

Response: We deposited all raw and processed data into Zenodo (DOI: 10.5281/zenodo.17170571) and updated the Data Availability section.

• Requirement: Competing Interests

Response: Updated to 'The authors have declared that no competing interests exist.'

• Requirement: ORCID

Response: The corresponding author’s ORCID iD has been validated in Editorial Manager.

We again thank the reviewer and editor for their constructive comments, which have greatly improved the clarity, rigor, and presentation of this work. We look forward to your favorable consideration.

Sincerely,

Yaser Norouzi, on behalf of all co-authors

• Additional Note on Comment 5 (Limitations):

A new subsection entitled 'Study Limitations' has been added to the Discussion. This section elaborates on baseline drift, weak absorbance signals, and potential ANN variability, as well as the need for larger and more diverse datasets to enhance generalizability.

---

## [Decision Letter · Decision Letter 1]

17 Oct 2025

Ultraviolet–Visible Spectral Characterization and ANN Modeling of Aqueous Sugar Solutions: Clinical and Environmental Perspectives

PONE-D-25-35917R1

Dear Dr. Norouzi,

We’re pleased to inform you that your manuscript has been judged scientifically suitable for publication and will be formally accepted for publication once it meets all outstanding technical requirements.

Kind regards,

Amitava Mukherjee, ME, Ph.D.

Academic Editor

PLOS ONE

Additional Editor Comments (optional):

Reviewers' comments:

Reviewer's Responses to Questions

**Comments to the Author**

Reviewer #1: All comments have been addressed

2. Is the manuscript technically sound, and do the data support the conclusions?

Reviewer #1: Yes

3. Has the statistical analysis been performed appropriately and rigorously?

Reviewer #1: Yes

4. Have the authors made all data underlying the findings in their manuscript fully available?

Reviewer #1: Yes

5. Is the manuscript presented in an intelligible fashion and written in standard English?

Reviewer #1: Yes

Reviewer #1: Congratulations on your efforts into this work, which is highly meets the readers interest in this domain.

**Do you want your identity to be public for this peer review?** For information about this choice, including consent withdrawal, please see our Privacy Policy

Reviewer #1: **Yes: ** Sathish Mohan

---

## [Editor Report · Acceptance letter]

PONE-D-25-35917R1

PLOS ONE

Dear Dr. Norouzi,

I'm pleased to inform you that your manuscript has been deemed suitable for publication in PLOS ONE. Congratulations! Your manuscript is now being handed over to our production team.

Kind regards,

on behalf of

Professor Dr. Amitava Mukherjee

Academic Editor

PLOS ONE